# Stochastic Optimization with Laggard Data Pipelines

**Naman Agarwal**
Google AI Princeton
Princeton, NJ 08540
namanagarwal@google.com

**Rohan Anil**
Google Research
Mountain View, CA 94043
rohananil@google.com

**Tomer Koren**
Tel Aviv University & Google
Tel Aviv, Israel
tkoren@tauex.tau.ac.il

**Kunal Talwar**
Apple[*]
Cupertino, CA 95014
ktalwar@apple.com

**Cyril Zhang**
Microsoft Research[†]
New York, NY 10012
cyrilzhang@microsoft.com

## Abstract

State-of-the-art optimization is steadily shifting towards massively parallel pipelines with extremely large batch sizes. As a consequence, CPU-bound preprocessing and disk/memory/network operations have emerged as new performance bottlenecks, as opposed to hardware-accelerated gradient computations. In this regime, a recently proposed approach is data echoing (Choi et al., 2019), which takes repeated gradient steps on the same batch while waiting for fresh data to arrive from upstream. We provide the first convergence analyses of "data-echoed" extensions of common optimization methods, showing that they exhibit provable improvements over their synchronous counterparts. Specifically, we show that in convex optimization with stochastic minibatches, data echoing affords speedups on the curvature-dominated part of the convergence rate, while maintaining the optimal statistical rate.

## 1 Introduction

Recent empirical successes in large-scale machine learning have been powered by massive data parallelism and hardware acceleration, with batch sizes trending beyond 10K+ images [46] or 1M+ tokens [9]. Numerous interdisciplinary sources [5, 12, 24, 33] indicate that the performance bottlenecks of contemporary deep learning pipelines can lie in many places other than gradient computation. In other words, since the initial breakthroughs in hardware-accelerated deep learning [14, 28, 37], GPUs (and TPUs, FPGAs, etc.) have become too fast for upstream data loaders and preprocessors to keep up with.

Choi et al. [13] propose *data echoing*, a simple and versatile way to improve training performance in this regime. Each stage of the data pipeline runs asynchronously, oblivious to whether its input has been refreshed upstream. In particular, the optimization algorithm may choose to take additional gradient steps before a minibatch is refreshed, rather than spend idle time waiting for more data. The authors present a large-scale proof-of-concept empirical study, and find that data echoing affords a 3.25× speedup in a network-bound ImageNet setting.

Some natural curiosities arise from this practice: *When might this overfit? How carefully should one adjust the step size of an echoed gradient? Does acceleration work?* A theoretical understanding of convergence guarantees for these data-echoed optimization algorithms is missing.

---

[*]Work performed while at Google Brain.

[†]Work performed while at Google AI Princeton and Princeton University.

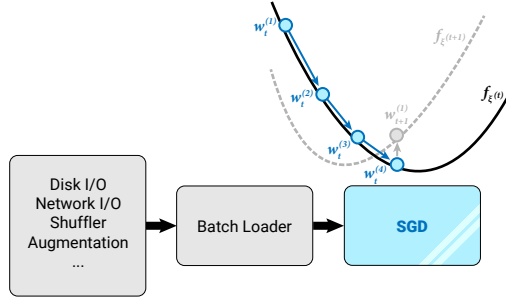

Figure 1: Schematic of data echoing, inspired by Choi et al. [13]. If the upstream data pipeline is $K = 4$ times slower than SGD, then SGD can potentially take that many steps on the same batch before the next one arrives.

|  | $T = 1$ | $T$ general |
|---|---|---|
| $K = 1$ | SGD | |
| $K$ small | Compute-bound ERM | Data echoing (Thm. 7) |
| $K$ large | Data-bound ERM | Approx-Prox [43] |
| $K \to \infty$ | Statistical ERM | Minibatch-Prox [43] |

Table 1: Regimes of echoing factor $K$ and number of batches $T$ which our analyses interpolates.

In this paper, we settle the issues of convergence and generalization for echoed gradient methods in convex optimization. We show that these methods can match the optimization performance of their non-stochastic counterparts, while achieving optimal statistical rates. As state-of-the-art batch sizes continue to grow, along with the distributed systems that enable them, we hope that this will provide a first theoretical grounding towards understanding the algorithmic and statistical challenges in these hardware-motivated optimization settings.

## 1.1 Technical contributions

Our model of data echoing is parameterized by the batch size $B$, the number of fresh i.i.d. batches $T$, and the *echoing factor* $K$, which is the number of gradient steps an algorithm can take on the (convex) loss on each batch. This reflects the hardware-determined setting where the data loader is at least $K$ times slower than the optimizer.

**Convergence in all data echoing regimes.** We first show that echoed SGD, with the correctly tuned step size, achieves a factor-$K$ speedup on the curvature term of the standard convergence rate, while keeping the optimal statistical term. Next, we develop an echoed method that is oblivious to the echoing factor $K$, getting the same rates for echoed SGD with an appropriately chosen proximal regularizer. Finally, we show that Nesterov's accelerated gradient descent, when echoed, achieves the optimal rates on quadratic losses. As a side contribution, we fix a small error in a technical lemma in [11], used in establishing the stability of AGD on quadratics. For general convex losses, we arrive at the same open question as these authors.

**Full interpolation between known regimes.** To set up notation, suppose that we go over $T$ batches of data, and perform $K$ echoed gradient steps for each batch. In the special case of $T = 1$ fresh batches, the problem becomes empirical risk minimization with a limited computational budget of $K$ gradient steps. When $K$ is small, the error is dominated by a *curvature* term, while for large enough $K$ this falls below the *statistical* term.

Motivated by the communication-limited setting, Wang et al. [43] focus on the case where $T$ is general and $K \to \infty$, analyzing the convergence of *exact* optimization of the prox-regularized minibatch loss. They develop a mild "approx-prox" guarantee when $K$ is large enough to enable an *exact+perturbation* analysis. Our analysis generalizes and strengthens these results, handling all values of $K$; Table 1 summarizes this discussion. When $B \to \infty$, the statistical problem disappears, and we recover the classical setting of full gradient descent with $KT$ oracle calls [8, 35].

| Algorithm | Standard | Data-echoed |
|---|---|---|
| SGD | $O\left(\dfrac{\beta D^2}{T} + \dfrac{\rho D}{\sqrt{BT}}\right)$ | $O\left(\dfrac{\beta D^2}{KT} + \dfrac{\rho D}{\sqrt{BT}}\right)$ |
| | (classical; see Lan [29]) | (Theorem 7) |
| Minibatch-Prox | $O\left(e^{-K/\kappa} + \dfrac{\rho D}{\sqrt{BT}}\right)$ | $O\left(\dfrac{\beta D^2}{KT} + \dfrac{\rho D}{\sqrt{BT}}\right)$ |
| | (Wang et al. [43]; $K$ large) | (Theorem 10) |
| Stochastic AGD | $O\left(\dfrac{\beta D^2}{T^2} + \dfrac{\rho D}{\sqrt{BT}}\right)$ | $O\left(\dfrac{\beta D^2}{K^2 T^2} + \dfrac{\rho D}{\sqrt{BT}}\right)$ |
| | (Lan [29]) | (Theorem 13; quadratics) |

Table 2: Single-step and *data-echoed* convergence rates of stochastic optimization algorithms studied in this paper. Notice that the optimization terms depend analogously on the total number of steps $KT$, and the statistical terms have optimal dependence on the total number of i.i.d. samples $BT$.

**Stability-based analysis.** We provide a modular proof framework for data echoing convergence bounds, based on uniform stability [7] and a potential-based notion of regret, which isolates the "bias" (curvature) and "variance" (generalization) components of the problem. This recipe (Theorem 4) can be used to sharpen bounds in more restricted settings, or analyze future data-echoed algorithms.

## 1.2 Motivation and context

It is well-known in the practice of GPU training that model parameter updates are not necessarily the performance bottleneck; this is why SSD storage is critical for pipelines on the scale of ImageNet [17]. For quantitative studies of I/O performance in deep learning, see [12, 45]. Many empirical advances have stemmed from innovations in data augmentation [15, 22, 41]. Unlike neural network training and inference, these data transformations can be highly sequential and/or heterogeneous, and must be done on CPU. Unlocking GPU parallelism for CPU-bound computations is often a significant engineering effort [16, 20, 27, 32].

Extremely large batch sizes have become the norm in training state-of-the-art models [4, 9, 18, 40, 46]. An overwhelming theme has been that *constant factors matter*; for example, memory-bound optimizers [2, 10, 39] care about factors of 2-3. When selecting hyperparameters in large-batch training setups, it is common to balance the curvature- and noise-dominated terms [26, 34, 38, 42]. This underscores the need to better understand the fine-grained dependences on $B, T$, and $K$, especially the resources at stake are on the scale of GPU-years.

The idea of repeated steps on a batch/workers has also been investigated in the context of federated learning [25], where a related concept is referred to as *local SGD* or *federated averaging*. There are two key distinctions: federated learning considers multiple copies of local SGD running on different workers, which synchronize intermittently through averaging; under the most simplified assumptions, each individual gradient step within a worker is taken on a fresh batch. While the improvements obtained in this recent and concurrent line of work (see [44] and references therein) bear resemblance to our bounds, we do not see a direct reduction in either direction. Indeed, due to the distinctions mentioned, getting similar improvements to the curvature term in federated learning is not possible beyond quadratics, as shown by [44]. Obtaining optimal rates for convex functions in the federated learning setting remains an interesting open problem.

## 1.3 The bias-variance problem in data echoing

As mentioned earlier, data echoing presents a natural tradeoff between the optimization gains from repeating gradient steps vs. the potential loss of generalization due to overfitting to stale batches. To understand this in detail, let us revisit the standard convergence guarantee for SGD on smooth functions:

$$\mathbb{E}[F(w_{\text{out}})] - F(w^*) \leq O\left(\frac{1}{T} + \frac{1}{\sqrt{BT}}\right).$$

We interpret the first term as a *bias* (*curvature*) term, which diminishes at a faster rate due to smoothness. The second term is the *variance* (*statistical*) term, which arises due to the stochasticity

in the data, and thus naturally scales as the inverse square root of the batch size. Viewing $B$ as fixed, the variance term is intrinsic to the data; therefore, we cannot expect data-echoing (or any algorithm) to give us improvements on that term for free. In fact, it is possible to make this term degrade, by overfitting on a batch. On the other hand, we can expect the bias term, which is governed by progress on the curvature of the underlying population loss, to decrease as we are given more echoing steps $K$. In light of this, the best analogous convergence rate one should hope to achieve in the data-echoing setting is

$$O\left(\frac{1}{KT} + \frac{1}{\sqrt{BT}}\right).$$

Our results establish exactly this rate for the data-echoed version of gradient descent. The data-echoed version of accelerated gradient descent is also shown to possess similar gains but with a faster rate of $K^2T^2$. The challenge is to prevent overfitting; obtaining such rates requires careful control (depending on $K$) of step sizes. Later, we alleviate this need via data-echoed proximal GD, whose parameters are independent of $K$.

### 1.4 Overview of techniques

All of our theorems follow the same analysis structure. In particular, we formalize a notion of *potential-bounded regret* (Definition 3), which connects an algorithm's function-value progress on a minibatch to a decrement on a certain potential function with respect to an arbitrary point. This potential function depends on the algorithm in question, but the key property is that it telescopes when summed over batches; this provides a fast rate on the bias term with respect to $T$.

The second piece of the analysis connects function-value decrease on a batch to the population objective via the notion of *uniform stability* (Definition 1). Note that the potential decrease scales inversely with $K$, whereas the stability constant increases with $K$ (unless a proximal regularizer is added). The key to maintaining the optimal statistical rate is to balance these terms via the choice of an appropriate step size. This type of algorithmic stability analysis has appeared various times in the literature [7, 11, 21]; we show here that it affords a way to analyze echoed gradient methods.

## 2 Preliminaries

### 2.1 Problem definition

Given a convex set $\mathcal{W} \subseteq \mathbb{R}^n$ and a domain $\Xi$ with a distribution $\mathcal{D}$, we consider the following stochastic convex optimization problem:

$$\underset{w \in \mathcal{W}}{\text{minimize}} \quad F(w) \overset{\text{def}}{=} \underset{\xi \sim \mathcal{D}}{\mathbb{E}}[f(w, \xi)]. \tag{1}$$

Here $f : \mathbb{R}^n \times \Xi \to \mathbb{R}$ is such that for any $\xi$, $f(\cdot, \xi)$ is convex, differentiable, $\rho$-Lipschitz, and $\beta$-smooth; i.e., for all $w, w' \in \mathcal{W}$,

$$f(w) - f(w') \leq \langle \nabla f(w'), w - w' \rangle + \frac{\beta}{2}\|w - w'\|^2.$$

When the minimizer exists, we define $w^* = \arg\min_{w \in \mathcal{W}} F(w)$. However, our results pertaining to optimality gaps $F(w) - F(w^*)$ hold for arbitrary $w^*$, encompassing the case when this minimizer does not exist. We further assume that we have access to an initial point $w_0$ with a bounded distance $D$ from the comparator; i.e., $\|w_0 - w^*\| \leq D$.

**Minibatch optimization.** We will work in the stochastic minibatch oracle model: at each time step $t$, we receive a new batch (of size $B$) examples $\boldsymbol{\xi}^{(t)} = \{\xi^{(t,i)}\}_{i=1}^B$ sampled i.i.d. from the distribution $\mathcal{D}$. For any batch of examples $\boldsymbol{\xi} = \{\xi^{(i)}\}$, we define the empirical objective on the batch as

$$\bar{f}_{\boldsymbol{\xi}}(w) \overset{\text{def}}{=} \frac{1}{|\boldsymbol{\xi}|} \sum_{i=1}^B f(w, \xi^{(i)}).$$

Throughout this paper, we will use **boldface $\boldsymbol{\xi}$** to denote a batch of $B$ examples, and unbolded $\xi$ to represent a single example in $\Xi$.

**Optimization algorithms.** We formalize a generic notion of optimization algorithms. Since these algorithms are called repeatedly by the data-echoing procedure, we will augment the output space of optimization algorithms with a notion of *state*, which it internally maintains and passes to the next

run of the same algorithm. Formally, an optimization algorithm is an iterative procedure which takes four arguments: an initial point $w_{\text{init}} \in \mathcal{W}$, an initial state $s_{\text{init}}$, the current batch $\boldsymbol{\xi}$ which determines the current objective $\bar{f}_{\boldsymbol{\xi}}$, and the number of steps $k$. The algorithm outputs a point $w_{\text{out}} \in \mathcal{W}$ and an output state $s_{\text{out}}$. In short, an algorithm $\mathcal{A}$ implements

$$(w_{\text{out}}, s_{\text{out}}) \leftarrow \mathcal{A}(w_{\text{init}}, s_{\text{init}}, \boldsymbol{\xi}, k).$$

We will suppress the notation of one or more of the arguments to $\mathcal{A}$ when they will be clear from the context, and write $f(\mathcal{A}(\cdot))$ as a shorthand for $f(w_{\text{out}})$, ignoring the auxiliary state $s_{\text{out}}$. Note that $w_{\text{out}}$ and $s_{\text{out}}$ are random variables, determined by the stochastic minibatch $\boldsymbol{\xi}$.

## 2.2 Algorithmic stability

**Definition 1** (Uniform stability). A deterministic[3] algorithm $\mathcal{A}$ is considered to be $\epsilon$-uniformly stable with respect to loss function $f : \mathcal{W} \times \Xi \to \mathbb{R}$ if, for two batches of data $\boldsymbol{\xi}, \boldsymbol{\xi}'$ differing in exactly one example, we have that

$$\sup_{\xi \in \Xi} | f(\mathcal{A}(\boldsymbol{\xi}), \xi) - f(\mathcal{A}(\boldsymbol{\xi}'), \xi) | \ \leq \ \epsilon.$$

The following is a well-known result connecting stability to generalization [7]. Here, we state a version taken from [21]:

**Theorem 2.** *If an algorithm $\mathcal{A}$ is $\epsilon$-uniformly stable, then it holds that*

$$\left| \mathop{\mathbb{E}}_{\boldsymbol{\xi} \sim \mathcal{D}^B} \left[ \bar{f}_{\boldsymbol{\xi}} \left( \mathcal{A}(\boldsymbol{\xi}) \right) - F \left( \mathcal{A}(\boldsymbol{\xi}) \right) \right] \right| \ \leq \ \epsilon.$$

## 3 The data echoing meta-algorithm

Given an minibatch optimization algorithm $\mathcal{A}$, its data-echoed extension is defined by Algorithm 1.

---

**Algorithm 1** Data echoing meta-algorithm

---

1: **Input:** Optimizer $\mathcal{A}$; initializer $w_{\text{init}} := w_0$; initial state $s_{\text{init}} := s_0$; number of inner steps $K$
2: **for** $t = 0, \ldots, T - 1$ **do**
3:      Receive a batch of examples $\boldsymbol{\xi}^{(t)} = \{\xi^{(t,i)}\}_{i=1}^{B}$.
4:      Execute $\mathcal{A}$ on $\boldsymbol{\xi}^{(t)}$ starting at $w_t$ for $K$ steps:    $(w_{t+1}, s_{t+1}) \leftarrow \mathcal{A}(w_t, s_t, \boldsymbol{\xi}^{(t)}, K)$.
5: **Output:** Average iterate $w_{\text{out}} := \frac{1}{T} \sum_{t=0}^{T-1} w_t$

---

### 3.1 Data-echoed algorithms

Using the framework of Algorithm 1, we introduce the data-echoed versions of three ubiquitous optimization algorithms. In [13], several types of data echoing are defined; we focus on what the authors call *batch echoing*.

**Data-echoed gradient descent.** We first formalize gradient descent in our optimization framework. The gradient descent procedure only contains the *fixed* learning rate as the state:

$$s_{\text{init}} = s_{\text{out}} := \{\eta\}.$$

The iterations defining the inner algorithm $\mathcal{A}$ are straightforward:

$$w_0 = w_{\text{init}}, \quad \{w_{j+1} = w_j - \eta \nabla \bar{f}_{\boldsymbol{\xi}}(w_j)\}_{j=0}^{K-1}, \quad w_{\text{out}} = w_K.$$

When Algorithm 1 is instantiated with this choice of $\mathcal{A}$, we call the overall procedure *data-echoed gradient descent*.

**Data-echoed proximal gradient descent.** The state of the proximally-regularized gradient descent procedure contains three variables: the fixed learning rate $\eta$, the prox parameter $\gamma$, and $w_{\text{pivot}}$, the center of the prox term:

$$s_{\text{init}} := \{\eta, \gamma, w_{\text{pivot}}\}.$$

We now define the proximal function

$$\bar{f}_{\text{prox}}(w) = \bar{f}_{\boldsymbol{\xi}}(w) + \frac{\gamma}{2} \|w - w_{\text{pivot}}\|^2.$$

The iterations proceed in same way as gradient descent, but on $\bar{f}_{\text{prox}}$:

$$w_0 = w_{\text{init}}, \quad \{w_{j+1} = w_j - \eta \nabla \bar{f}_{\text{prox}}(w_j)\}_{j=0}^{K-1}, \quad w_{\text{out}} = w_K.$$

The output returned is $s_{\text{out}} = \{\eta, \gamma, \frac{1}{K} \sum_{j=0}^{K-1} w_j\}$. This particular choice of returning the average iterate as the next $w_{\text{pivot}}$ simplifies our analysis. With this choice of $\mathcal{A}$, this overall procedure will be called *data-echoed proximal gradient descent*.

**Data-echoed accelerated gradient descent.** The state space for accelerated gradient consists of a step size $\eta$, an initial momentum vector $d$, and a momentum scale factor $\lambda$; thus $s_{\text{init}} = \{\eta, d, \lambda\}$. Define the following scalar sequences with $\lambda_0 = \lambda$:

$$\lambda_{j+1}^2 - \lambda_{j+1} = \lambda_j^2, \qquad \gamma_{j+1} = \frac{\lambda_j - 1}{\lambda_{j+1}}.$$

The updates now follow the progression as in Nesterov's acceleration [36]:

$$w_0 = w_{\text{init}}, \quad d_0 = d, \quad w_{j+1} = (w_j + d_j) - \eta \nabla \bar{f}_{\xi}(w_j + d_j), \quad d_{j+1} = \gamma_{j+1}(w_{j+1} - w_j).$$

Finally, the outputs are given by $s_{\text{out}} = \{\eta, d_K, \lambda_K\}, w_{\text{out}} = w_K$.

With this choice of $\mathcal{A}$, we refer to the overall procedure as *data-echoed accelerated gradient descent*.

# 4 Convergence analyses of echoed methods

We will analyze the data-echoing algorithms by separating their optimization properties from their stability properties. For the latter, we use the standard notion of uniform stability, as defined earlier. For the optimization part, we use a notion of potential-bounded regret, which we define next.

**Definition 3** (Potential-bounded regret). We say that an algorithm $\mathcal{A}$ has *potential-bounded regret* with potential function $V_{\mathcal{A}}$ if given a $\beta$ smooth convex function $f$ on a domain $\mathcal{W}$ and a starting point $w_{\text{init}}$, $\mathcal{A}$ produces a point $w_{\text{out}}$ such that for all $w^* \in \mathcal{W}$, it holds that

$$f(w_{\text{out}}) - f(w^*) \leq V_{\mathcal{A}}(w_{\text{init}}, s_{\text{init}}, w^*) - V_{\mathcal{A}}(w_{\text{out}}, s_{\text{out}}, w^*).$$

This inequality is a fundamental lemma in the standard analysis of mirror descent (see [6], or Section B.2 from [1]), but we extend it to *nested stateful algorithms* instead of a single step. For the echoed algorithms we analyze in this work, squared Euclidean norms will be suitable potentials.

We state and prove our main generic theorem below:

**Theorem 4.** *Let $\mathcal{A}$ be an $\epsilon$-uniformly stable algorithm. Furthermore, suppose $\mathcal{A}$ has the potential-bounded regret property with respect to $V_{\mathcal{A}}$. Then, for any $w^* \in \mathcal{W}$, Algorithm 1 with inner algorithm $\mathcal{A}$ satisfies*

$$\mathbb{E}[F(w_{\text{out}})] - F(w^*) \leq \frac{V_{\mathcal{A}}(w_0, s_0, w^*) - \mathbb{E}[V_{\mathcal{A}}(w_T, s_T, w^*)]}{T} + \epsilon.$$

*Proof.* From the potential-bounded regret property of the algorithm $\mathcal{A}$, we get that

$$\bar{f}_{\xi^{(t)}}(w_{t+1}) - \bar{f}_{\xi^{(t)}}(w^*) \leq V_{\mathcal{A}}(w_t, s_t, w^*) - V_{\mathcal{A}}(w_{t+1}, s_{t+1}, w^*).$$

Let $\mathbb{E}_t[\cdot]$ denote the expectation conditioned on all randomness in the minibatches up to (and including) time $t$. We now get from the uniform stability of $\mathcal{A}$ that

$$\mathbb{E}[F(w_{t+1})] = \mathbb{E}_{t-1} \mathbb{E}_{\xi^{(t)}}[F(w_{t+1})] \leq \mathbb{E}_{t-1}\left[\mathbb{E}_{\xi^{(t)}}[\bar{f}_{\xi^{(t)}}(w_{t+1})] + \epsilon\right].$$

Thus we have

$$\mathbb{E}[F(w_{t+1})] - F(w^*) \leq \mathbb{E}_{t-1} \mathbb{E}_{\xi^{(t)}}[\bar{f}_{\xi^{(t)}}(w_{t+1}) - \bar{f}_{\xi^{(t)}}(w^*)] + \epsilon$$

$$\leq \mathbb{E}_{t-1} \mathbb{E}_{\xi^{(t)}}[V_{\mathcal{A}}(w_t, s_t, w^*) - V_{\mathcal{A}}(w_{t+1}, s_{t+1}, w^*)] + \epsilon$$

$$\leq \mathbb{E}[V_{\mathcal{A}}(w_t, s_t, w^*)] - \mathbb{E}[V_{\mathcal{A}}(w_{t+1}, s_{t+1}, w^*)] + \epsilon.$$

Summing the above over time and using the convexity of $F$ gives us that

$$\mathbb{E}[F(w_{\text{out}})] - F(w^*) \leq \sum_{t=0}^{T-1} \frac{\mathbb{E}[F(w_{t+1})] - F(w^*) + \epsilon}{T}$$

$$\leq \frac{V_{\mathcal{A}}(w_0, s_0, w^*) - \mathbb{E}[V_{\mathcal{A}}(w_T, s_T, w^*)]}{T} + \epsilon. \qquad \blacksquare$$

In the rest of the section, we present various applications of our main data echoing theorem. In each case, we will consider a standard algorithm, derive its stability and potential bounded regret properties, then use Theorem 4 to derive the convergence rate for its echoed version. All regret proofs can be found in Appendix A, and stability proofs in Appendix B; the corresponding convergence rates for the echoed algorithms are proven in Appendix C.

## 4.1 Echoed gradient descent

We begin by establishing the following properties of gradient descent. In the rest of the theorem and lemma statements in this section $w^*$ is an arbitrary point in $\mathcal{W}$.

**Lemma 5** (Potential-bounded regret for GD). *Let $f$ be a $\beta$-smooth convex function. Then $K$ steps of gradient descent on $f$, with a step size $\eta \leq 1/\beta$, satisfies the potential-bounded regret property with $V(w, s, w^*) := \frac{1}{2}\|w - w^*\|^2$:*

$$f(w_{\text{out}}) - f(w^*) \leq \frac{1}{\eta K}\left(\frac{\|w_{\text{init}} - w^*\|^2}{2} - \frac{\|w_{\text{out}} - w^*\|^2}{2}\right).$$

**Lemma 6** (Stability of GD). *For a $\beta$-smooth function $f$, and any $0 \leq \eta \leq 1/\beta$, gradient descent on $f$, run with step size $\eta$ for $K$ steps, is $\epsilon$-uniformly stable with $\epsilon = 2\eta K \rho^2 / B$.*

Combining Lemmas 5 and 6, we conclude the following convergence bound for data-echoed GD:

**Theorem 7** (Data-echoed GD). *$T$ outer steps of data-echoed gradient descent, with a step size of $\eta = \min\left\{\frac{1}{\beta}, \frac{\rho}{KD}\sqrt{\frac{B}{T}}\right\}$ and $K$ internal steps, produces a point $w_{out}$ satisfying*

$$\mathbb{E}[F(w_{\text{out}})] - F(w^*) \leq \frac{\beta D^2}{2KT} + \frac{2\rho D}{\sqrt{BT}}.$$

## 4.2 Echoed proximal gradient descent

For proximal GD, we derive the following bounds on potential-bounded regret and stability:

**Lemma 8.** *Let $f$ be a $\beta$-smooth convex function. Consider the potential function*

$$V(w, \{\eta, \gamma, w_{\text{pivot}}\}, w^*) = \frac{\|w - w^*\|^2}{2\eta K} + \frac{\gamma\|w - w_{\text{pivot}}\|^2}{2}.$$

*Then $K$-step proximal gradient descent, with step-size $\eta \leq 1/(\beta + \gamma)$ has regret bounded by*

$$f(w_{\text{out}}) - f(w^*) \leq V(w_{\text{out}}, s_{\text{out}}, w^*) - V(w_{\text{init}}, s_{\text{init}}, w^*).$$

**Lemma 9** (Stability of prox-GD). *For a $\beta$-smooth function $f$, any $\lambda \geq 0$ and any $0 \leq \eta \leq 1/(\beta+\lambda)$, $K$ steps of proximal gradient descent are $\epsilon$-uniformly stable with $\epsilon = \frac{2\rho^2}{B\gamma}\left(1 - (1 - \eta\gamma)^K\right)$.*

The proofs of both lemmas are included in the the supplementary material. Combining Lemmas 8 and 9, we get the following guarantee on the performance of data-echoed prox-GD (proof included in the supplementary material):

**Theorem 10** (Data-echoed prox-GD). *$T$ outer steps of echoed gradient descent, with a prox parameter of $\gamma = \frac{\rho}{D}\sqrt{\frac{T}{B}}$, step size $\eta = \frac{1}{\beta+\gamma}$, and $K$ internal steps, produces a point $w_{\text{out}}$ satisfying*

$$\mathbb{E}[F(w_{\text{out}})] - F(w^*) \leq \sqrt{1 + \frac{1}{K}} \cdot \frac{2\rho\|w_{\text{init}} - w^*\|}{\sqrt{BT}} + \frac{\beta\|w_{\text{init}} - w^*\|^2}{2KT}.$$

Note that using this algorithm, the correct choice of step size $\eta$ no longer depends on the echoing factor $K$. In fact, even if $K$ varies across the execution of the proximal algorithm, a straightforward extension of our analysis shows that proximal gradient descent can achieve $\sum_t K_t$ instead of the $KT$ factor in the denominator of the bias term. This resilience to indeterminate echoing factors is especially appealing for the motivating setting of asynchronous pipelines.

### 4.3 Echoed accelerated gradient descent

For the case of Nesterov's accelerated gradient descent, we consider a slightly modified version of our data-echoing meta-procedure. This arises from the fact that even the stochastic setting of accelerated gradient descent, algorithms output the final iterate and not the average iterate. The resulting slightly modified procedure is outlined in Algorithm 2.

---

**Algorithm 2** Data-echoing meta-algorithm (final iterate)

1: **Input:** Optimizer $\mathcal{A}$; initializer $w_{\text{init}} := w_0$; initial state $s_{\text{init}} := s_0$; number of inner steps $K$.
2: **for** $t = 0, \dots, T - 1$ **do**
3:     Receive a batch of examples $\boldsymbol{\xi}^{(t)} = \{\xi^{(t,i)}\}_{i=1}^{B}$.
4:     Execute $\mathcal{A}$ on $\boldsymbol{\xi}^{(t)}$ starting at $w_t$ for $K$ steps:    $w_{t+1}, s_{t+1} \leftarrow \mathcal{A}(w_t, s_t, \boldsymbol{\xi}^{(t)}, K_t)$.
5: **Output:** Final iterate $w_{\text{out}} := w_T$

---

We also add a slight extension to our potential-based regret abstraction:

**Lemma 11** (Potential-bounded regret for AGD). *Let $f$ be a $\beta$-smooth convex function. Running accelerated gradient descent for $K$ steps, with a step size $\eta \leq 1/\beta$, gives the regret bound*

$$(\lambda_{\text{out}}^2 - \lambda_{\text{out}})(f(w_{\text{out}}) - f(w)) - (\lambda_{\text{init}}^2 - \lambda_{\text{init}})(f(w_{\text{init}}) - f(w))$$

$$\leq \frac{1}{2\eta}(\|w_{\text{init}} + \lambda_{\text{init}}d_{\text{init}} - w\|^2 - \|w_{\text{out}} + \lambda_{\text{out}}d_{\text{out}} - w\|^2).$$

Further, to bound the stability, we note the following lemma which was essentially proved in [11]. Since we believe there is a small typo in the main argument in the original presentation of the proof, we provide an alternate derivation in the supplementary material.

**Lemma 12** (Stability of AGD). *Suppose that $f$ is a $\beta$-smooth convex quadratic function of $w$ for any $\xi$. Then, for any $0 \leq \eta \leq 1/\beta$ and initial state $s_{\text{init}}$, $K$ steps of accelerated gradient descent are $\epsilon$-uniformly stable with $\epsilon = O(\eta\rho^2 K^2/B)$.*

Combining Lemmas 11 and 12 we obtain the following guarantee for data-echoed AGD:

**Theorem 13.** *Suppose $f$ is a convex quadratic in $w$, for all $\xi$. Then, $T$ outer steps of echoed AGD, with echoing factor $K$ and step size $\eta = \Theta(\min\{\frac{1}{\beta}, \frac{\rho}{K^2 D \sqrt{B/T^{3/2}}}\})$, produces a point $w_{out}$ satisfying*

$$\mathbb{E}[F(w_{\text{out}})] - F(w^*) = O\left(\frac{\beta\|w_0 - w^*\|^2}{K^2 T^2} + \frac{\rho\|w_0 - w^*\|}{\sqrt{BT}}\right).$$

## 5 Experiments

We demonstrate numerical experiments on convex machine learning benchmarks. This acts as a validation of our theoretical findings, as well as a way to examine "beyond worst-case" phenomena not captured by our minimax convergence guarantees. This can be seen as a combination of the experiments of Figures 4-6 in [13], where we have exchanged the state-of-the-art setting for a more robust one, allowing for a closer dissection of the bias-variance decomposition.

**Methodology.** We consider two logistic regression problems as a benchmark, the scaled CoverType dataset from the UCI repository [19], and MNIST [30]. We record the number of iterations (including as well as excluding the data-echoing iterations) needed for SGD to reach within 1% of the optimum training loss, as we increase the echoing factor $K$, and thus decrease the *rate* of fresh independent samples usable by SGD. For each choice of $(B, K)$, we tune a constant learning rate by grid search, to minimize this time. All details can be found in the supplementary material.

**Results and discussion.** Figures 2 and 3 show our findings. As batch size $B$ increases, there is a phase transition from a variance-dominated regime (the $O(\rho D/\sqrt{BT})$ term in our analysis is larger) to a bias-dominated regime (the $O(\beta D/KT)$ term is larger). In the former regime, data-echoed SGD saturates on the stale data, and the optimal learning rate scales inversely with $K$, as predicted by the theory. In the latter regime, echoing attains a nearly embarrassingly-parallel speedup, and the optimal learning rate is close to constant. These experiments provide an end-to-end example of how the bias-variance decomposition can help to understand and diagnose the benefits and limitations of data-echoed algorithms.

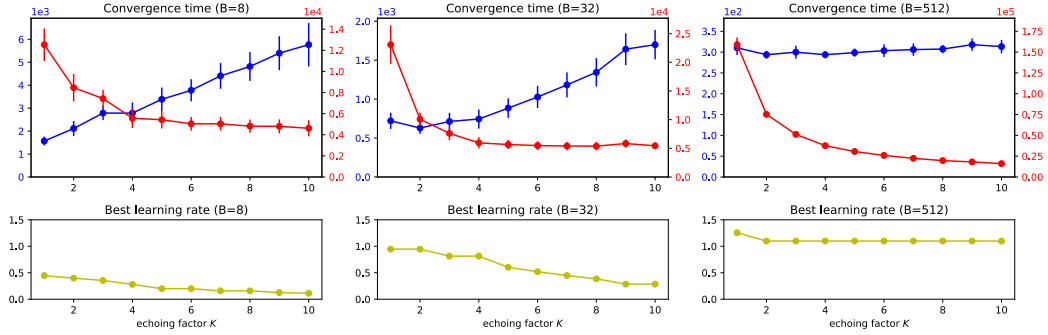

Figure 2: Convergence times as a function of echoing factor $K$, for logistic regression on the CoverType dataset. Learning rates (yellow) are tuned for each $(B, K)$ to minimize convergence times. Convergence times are presented in number of SGD steps $KT$ (blue), as well as number of independent samples consumed $BT$ (red). Note that the red curves reflect wall-clock time for data-echoing when the data loader is $K$ times slower than the optimizer. As batch size $B$ increases, we move from the noise-dominated regime (red curve plateaus) to the curvature-dominated regime (blue curve plateaus).

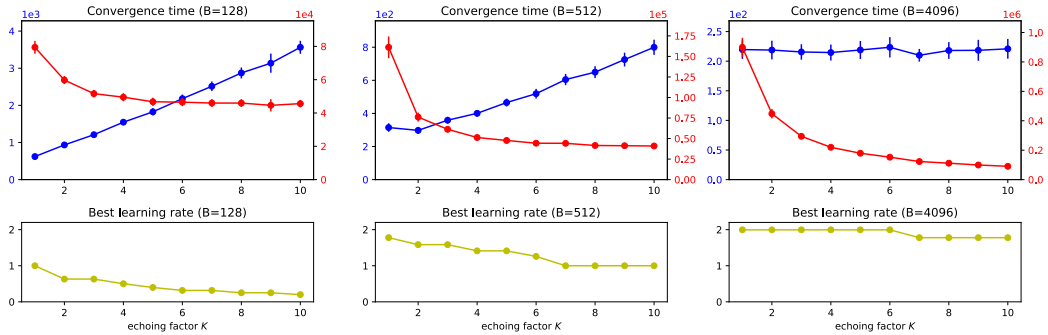

Figure 3: Convergence times, as in Figure 2, for logistic regression on the MNIST dataset. Note that the phase transition from noise-dominated to curvature-dominated regimes happens in a batch size range commonly used in deep learning benchmarks with this dataset.

**A note on deep neural nets.** Our theoretical setting was originally motivated by hardware constraints most frequently encountered in the massively parallel training of deep neural networks. Beyond the convex setting, we note that the experimental design problem become significantly more challenging. Some potential confounds include the learning rate choice affecting the generalization gap [23], and counterintuitive interactions between learning rate and batch normalization [3, 31]. In [13], the authors study the *end-to-end* performance gains of data echoing. Indeed, those experiments need many tweaks (like *example-wise* echoing, data re-augmentation, and individually tuned momentum and learning rate schedules) to obtain their most impressive speedups.

# 6   Conclusion

We have established first theoretical analysis in the nascent field of optimization algorithms for asynchronous data pipelines, where we have found that gradient descent and well-known variants can be adapted to resist overfitting to stale data. An immediate open problem is to develop a corresponding theory for local convergence and saddle point avoidance in the non-convex setting. This work provides further motivation to show the $O(\eta \rho^2 K^2 / B)$-uniform stability of AGD for smooth convex functions, which was conjectured in [11] with different motives. More broadly, we hope that the design and analysis of algorithms in optimization for machine learning can derive fruitful inspiration from nascent hardware considerations, like those that motivated this work.

## Broader Impact

This work is theoretical in nature, and is concerned with the very general framework of stochastic optimization. As such, there are no foreseen ethical or societal consequences for the research presented herein. We hope that by providing theoretical groundwork and algorithmic techniques for efficient large-scale optimization in settings informed by modern developments in optimization, works like this one will contribute to alleviating the steep resource and energy costs of large-scale machine learning.

## Acknowledgments and Disclosure of Funding

We are grateful to George Dahl and Yoram Singer for helpful discussions. CZ was supported by Elad Hazan's NSF grants IIS-1523815 and CCF1704860, and a Google internship.

## Footnotes

[3]A similar definition exists for randomized algorithms [7]. In this work, we focus on deterministic algorithms.

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
