[Supplementary Material]

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

# A Proofs for potential-bounded regret lemmas

In this section we provide the proofs of Lemmas 5, 8 and 11, which concern potential-bounded regret.

*Proof of Lemma 5.* To remind the reader, a step of gradient descent with step-size $\eta$ is given by

$$w_{j+1} = w_j - \eta \nabla f(w_j),$$

with $w_0 := w_{\text{init}}$ and $w_{\text{out}} := w_K$. Further fix an arbitrary $w^* \in \mathcal{W}$. Using the definition of $w_{j+1}$ and convexity we get that,

$$\begin{aligned}
f(w_j) - f(w^*) &\leq \nabla f(w_j)(w_j - w^*) \\
&\leq \frac{1}{2\eta}\left(\|w_j - w^*\|^2 - \|w_{j+1} - w^*\|^2\right) + \frac{\eta}{2}\|\nabla f(w_j)\|^2.
\end{aligned} \tag{2}$$

Furthermore, using $\beta$-smoothness we get

$$\begin{aligned}
f(w_{j+1}) - f(w_j) &\leq \nabla f(w_j)(w_{j+1} - w_j) + \frac{\beta}{2}\|w_{j+1} - w_j\|^2 \\
&\leq -\eta(1 - \tfrac{1}{2}\eta\beta)\|\nabla f(w_j)\|^2.
\end{aligned}$$

Therefore for $0 \leq \eta \leq 1/\beta$, we have that

$$\begin{aligned}
\|\nabla f(w_j)\|^2 &\leq \frac{1}{\eta(1 - \tfrac{1}{2}\eta\beta)}\left(f(w_j) - f(w_{j+1})\right) \\
&\leq \frac{2}{\eta}\left(f(w_j) - f(w_{j+1})\right).
\end{aligned} \tag{3}$$

Collecting Eqs. (2) and (3), summing and rearranging, we obtain

$$\sum_{t=0}^{K-1}\left(f(w_{j+1}) - f(w^*)\right) \leq \frac{1}{2\eta}\left(\|w_0 - w^*\|^2 - \|w_K - w^*\|^2\right).$$

Finally, observe that Eq. (3) also implies $f(w_K) \leq f(w_j)$ for all $1 \leq j \leq K$, which now gives the lemma. ∎

*Proof of Lemma 8.* Fix an arbitrary $w^* \in \mathcal{W}$. Using the definition of $w_{j+1}$ and the $\lambda$ strong-convexity of $f_{\text{prox}}$ we get that,

$$\begin{aligned}
f_{\text{prox}}(w_j) - f_{\text{prox}}(w^*) &\leq \nabla f_{\text{prox}}(w_j)(w_j - w^*) - \frac{\gamma}{2}\|w_j - w^*\|^2 \\
&\leq \frac{1}{2\eta}\left(\|w_j - w^*\|^2 - \|w_{j+1} - w^*\|^2\right) + \frac{\eta}{2}\|\nabla f_{\text{prox}}(w_j)\|^2 - \frac{\gamma}{2}\|w_j - w^*\|^2.
\end{aligned} \tag{4}$$

Furthermore, using the $(\beta + \gamma)$-smoothness of $f_{\text{prox}}$ we get

$$\begin{aligned}
f_{\text{prox}}(w_{j+1}) - f_{\text{prox}}(w_j) &\leq \nabla f_{\text{prox}}(w_j)(w_{j+1} - w_j) + \frac{\beta + \gamma}{2}\|w_{j+1} - w_j\|^2 \\
&= -\eta(1 - \tfrac{1}{2}\eta(\beta + \gamma))\|\nabla f_{\text{prox}}(w_j)\|^2.
\end{aligned}$$

Therefore, for $0 \leq \eta \leq 1/(\beta + \gamma)$, we have that

$$\begin{aligned}
\|\nabla f_{\text{prox}}(w_j)\|^2 &\leq \frac{1}{\eta(1 - \tfrac{1}{2}\eta(\beta + \gamma))}\left(f_{\text{prox}}(w_j) - f_{\text{prox}}(w_{j+1})\right) \\
&\leq \frac{2}{\eta}\left(f_{\text{prox}}(w_j) - f_{\text{prox}}(w_{j+1})\right).
\end{aligned} \tag{5}$$

Collecting Eqs. (4) and (5), summing and rearranging, we obtain

$$\begin{aligned}
\frac{1}{K}\sum_{t=0}^{K-1}\left(f_{\text{prox}}(w_{j+1}) - f_{\text{prox}}(w^*)\right) &\leq \frac{1}{2\eta K}\left(\|w_0 - w^*\|^2 - \|w_K - w^*\|^2\right) - \sum_{j=0}^{K-1}\frac{\gamma}{2K}\|w_j - w^*\|^2 \\
&\leq \frac{1}{2\eta K}\left(\|w_0 - w^*\|^2 - \|w_K - w^*\|^2\right) - \frac{\gamma}{2}\left\|\frac{\sum_j w_j}{K} - w^*\right\|^2 \\
&\leq \frac{1}{2\eta K}\left(\|w_0 - w^*\|^2 - \|w_K - w^*\|^2\right) - \frac{\gamma}{2}\|s_{\text{out}} - w^*\|^2.
\end{aligned}$$

Finally, observe that Eq. (5) also implies $f_{\text{prox}}(w_K) \leq f_{\text{prox}}(w_j)$ for all $j$. Therefore we have that

$$f(w_{\text{out}}) - f(w^*) - \frac{\gamma}{2}\|w^* - s_{\text{init}}\|^2 \leq \frac{1}{K}\sum_{t=0}^{K-1}\left(f_{\text{prox}}(w_{j+1}) - f_{\text{prox}}(w^*)\right)$$

$$\leq \frac{1}{2\eta K}\left(\|w_0 - w^*\|^2 - \|w_K - w^*\|^2\right) - \frac{\gamma}{2}\|s_{\text{out}} - w^*\|^2,$$

This concludes the lemma. ∎

*Proof of Lemma 11.* Let $x_{j+1} \stackrel{\text{def}}{=} w_j + d_j$. Fix an arbitrary $w^* \in \mathcal{W}$ and define $h(w) \stackrel{\text{def}}{=} f(w) - f(w^*)$. We will now collect a host of inequalities that will be useful. First by smoothness and the choice of $\eta$ we have

$$f(w_{j+1}) - f(x_{j+1}) \leq -\frac{\eta}{2}\|\nabla f(x_{j+1})\|^2.$$

Further, by convexity we have

$$f(x_{j+1}) - f(w_j) \leq \nabla f(x_{j+1})^\top d_j;$$
$$f(x_{j+1}) - f(w^*) \leq \nabla f(x_{j+1})^\top(w_j + d_j - w^*).$$

Adding the above we get

$$h(w_{j+1}) - h(w_j) \leq -\frac{\eta}{2}\|\nabla f(x_{j+1})\|^2 + \nabla f(x_{j+1})^\top d_j; \tag{6}$$

$$h(w_{j+1}) \leq -\frac{\eta}{2}\|\nabla f(x_{j+1})\|^2 + \nabla f(x_{j+1})^\top(w_j + d_j - w^*). \tag{7}$$

Furthermore, note that

$$\lambda_j\|\eta\nabla f(x_{j+1})\|^2 + 2\eta\nabla f(x_{j+1})^\top(w_j + \lambda d_j - w)$$

$$= \frac{1}{\lambda_j}\left(\|w_j + \lambda_j d_j - w^* + \lambda_j\eta\nabla f(w_{j+1})\|^2 - \|w_j + \lambda_j d_j - w^*\|^2\right)$$

$$= \frac{1}{\lambda_j}\left(\|w_{j+1} + \lambda_{j+1}d_{j+1} - w^*\|^2 - \|w_j + \lambda_j d_j - w^*\|^2\right). \tag{8}$$

Adding $(\lambda_j - 1)$ times Eq. (6), 1 times Eq. (7) and $(-1/2\eta)$ times Eq. (8) gives us

$$\lambda_j^2 h(w_{j+1}) - (\lambda_j^2 - \lambda_j)h(w_j) \leq \frac{1}{2\eta}(u_j - u_{j+1}),$$

where $u_j = \|w_j + \lambda_j d_j - w^*\|^2$. Summing this over time we get

$$(\lambda_K^2 - \lambda_K)h(w_K) - (\lambda_0^2 - \lambda_0)h(w_0) = \lambda_{K-1}^2 h(w_K) - (\lambda_0^2 - \lambda_0)h(w_0)$$

$$\leq \frac{1}{2\eta}(\|w_0 + \lambda_0 d_0 - w^*\|^2 - \|w_K + \lambda_K d_K - w^*\|^2)$$

which finishes the proof.

∎

,

# B    Stability proofs

In this section, we prove the bounds on the stability of the respective algorithms (Lemmas 6, 9 and 12). Our general recipe for showing stability of various algorithms would be to show that the points visited by the iterative algorithms themselves do not differ by much. To this end, note that since $f$ is Lipschitz, we have that

$$\sup_{\xi \in \Xi}|f(\mathcal{A}(\boldsymbol{\xi}), \xi) - f(\mathcal{A}(\boldsymbol{\xi}'), \xi)| \leq \rho\|\mathcal{A}(\boldsymbol{\xi}) - \mathcal{A}(\boldsymbol{\xi}')\|. \tag{9}$$

Thus, it is sufficient to show to bound $\mathcal{A}(\boldsymbol{\xi}) - \mathcal{A}(\boldsymbol{\xi}')$, which is what we do next.

*Proof of Lemma 6.* For simplicity of presentation we assume that the Hessian of $f$ is a continuous function. The more general case can be derived by following the arguments in [21].

Let $w_j^{\boldsymbol{\xi}}$ and $w_j^{\boldsymbol{\xi'}}$ denote the points generated by gradient descent on $\boldsymbol{\xi}$ and $\boldsymbol{\xi'}$ respectively. Further define

$$\Delta w_j := w_j^{\boldsymbol{\xi'}} - w_j^{\boldsymbol{\xi}};$$

$$\nabla \bar{f}_{\boldsymbol{\xi}}(\Delta w_j) := \nabla \bar{f}(w_j^{\boldsymbol{\xi'}}) - \nabla \bar{f}(w_j^{\boldsymbol{\xi}}).$$

Therefore via the mean value theorem, we can write this as

$$\nabla \bar{f}_{\boldsymbol{\xi}}(\Delta w_j) = H_j(\Delta w_j)$$

for some $H_j$ along the line segment from $w_j^{\boldsymbol{\xi}}$ to $w_j^{\boldsymbol{\xi'}}$. It is now easy to see

$$\Delta w_{j+1} = (I - \eta H_j)\Delta w_j + \frac{\eta}{B}(\nabla f(w_j^{\boldsymbol{\xi'}}, \xi') - \nabla f(w_j^{\boldsymbol{\xi'}}, \xi)).$$

Noting that $0 \preceq I - \eta H_j \preceq I$(by the choice of $\eta$) and that $\|\nabla f\| \leq \rho$, we have that

$$\|\Delta w_{j+1}\| \leq \|\Delta w_j\| + \frac{2\eta\rho}{B}.$$

The proof is now finished by using Eq. (9). ∎

*Proof of Lemma 9.* The proof follows the exact same structure as in the proof of Lemma 6, except at the end where since the prox function is $\lambda$ strongly convex we get that

$$0 \preceq I - \eta H_j \preceq (1 - \eta\gamma)I.$$

Replacing this gives us

$$\|\Delta w_{j+1}\| \leq (1 - \eta\gamma)\|\Delta w_j\| + \frac{2\eta\rho}{B}.$$

Unrolling the above over $0 \leq j \leq K - 1$ and using Eq. (9) gives the result. ∎

*Proof of Lemma 12.* As mentioned before the proof follows exactly along the lines of the proof of Theorem 11 in [11]. It can easily be seen from the original proof that the presence of an initial momentum term $d_0$ (which is assumed to be 0 in the original proof) makes no difference to the arguments. Furthermore starting the $\lambda$ sequence from $\lambda_j$ also does not make any difference to the proof, as it only requires $-1 \leq \gamma_j \leq 0$ which our sequence also continues to satisfy irrespective of the choice of $\lambda_0$.

We believe that there is a small typo in the main argument of the original proof in Lemma 20 in [11]. We fix the slight indexing error of the argument. In particular, the proof boils down to showing the following lemma.

**Lemma 14** (Lemma 20, [11]). *Suppose*

$$H_i = \begin{bmatrix} (1 - \gamma_i)h & \gamma_i h \\ 1 & 0 \end{bmatrix}$$

*where $h \in [0, 1]$ and $\gamma_i \in (-1, 1)$. Then, for all $t \in \mathbb{N}$,*

$$\left\| \prod_{i=1}^{j} H_i \right\|_2 \leq 2(j + 1).$$

The proof of the lemma proceeds by analyzing the cases when $\mathbf{y}_i \in \{-1, 1\}$. The only case where we differ from the presented proof is when $\gamma_i = -1$. In this case, we have

$$H_i = H := \begin{bmatrix} 2h & -h \\ 1 & 0 \end{bmatrix},$$

and we need to bound the operator norm of the powers of this matrix for all $h \in [0, 1]$.

**Lemma 15.** *For any $n \geq 1$, we have*

$$H^{2n} = h^n \begin{bmatrix} U_{2n}(\sqrt{h}) & -\sqrt{h} \cdot U_{2n-1}(\sqrt{h}) \\ \frac{U_{2n-1}(\sqrt{h})}{\sqrt{h}} & -U_{2n-2}(\sqrt{h}) \end{bmatrix},$$

*and*

$$H^{2n+1} = h^n \begin{bmatrix} \sqrt{h}U_{2n+1}(\sqrt{h}) & -h \cdot U_{2n}(\sqrt{h}) \\ U_{2n}(\sqrt{h}) & -\sqrt{h}U_{2n-1}(\sqrt{h}) \end{bmatrix},$$

*where $U_n(\cdot)$ is the n-th Chebyshev polynomial of the second kind.*

*Proof.* We begin by proving the identity for even powers $2n$, by induction. The base case $n = 1$ holds by manual computation, noting the following facts:

$$H^2 = h \begin{bmatrix} 4h - 1 & -2h \\ 2 & -1 \end{bmatrix},$$

$$U_0(x) = 1, \quad U_1(x) = 2x, \quad U_2(x) = 4x^2 - 1.$$

Next, we prove the inductive step, showing that the identity for $2n$ implies the same for $2n + 2$. Below, we substitute $r := \sqrt{h}$ for clarity:

$$H^{2n+2} = h \cdot \begin{bmatrix} 4h - 1 & -2h \\ 2 & -1 \end{bmatrix} H_i^{2n}$$

$$= h^{n+1} \begin{bmatrix} 4r^2 - 1 & -2r^2 \\ 2 & -1 \end{bmatrix} \begin{bmatrix} U_{2n}(r) & -rU_{2n-1}(r) \\ \frac{U_{2n-1}(r)}{r} & -U_{2n-2}(r) \end{bmatrix},$$

Computing each entry of the matrix product, and applying the recurrence $U_{n+1}(r) = 2rU_n(r) - U_{n-1}(r)$:

$$[H^{2n+2}]_{11} = h^{n+1} \left( (4r^2 - 1)U_{2n}(r) - 2rU_{2n-1}(r) \right)$$

$$= h^{n+1} \left( 2rU_{2n+1}(r) - U_{2n}(r) \right) = h^{n+1} U_{2n+2}(r),$$

$$[H^{2n+2}]_{12} = -h^{n+1} \left( r(4r^2 - 1)U_{2n-1}(r) - 2r^2 U_{2n-2}(r) \right)$$

$$= -h^{n+1} r \left[ H^{2n-1} \right]_{11} = h^{n+1} r U_{2n+1}(r),$$

$$[H^{2n+2}]_{21} = h^{n+1} \cdot \frac{2rU_{2n}(r) - U_{2n-1}(r)}{r} = h^{n+1} \frac{U_{2n+1}(r)}{r},$$

$$[H^{2n+2}]_{22} = -h^{n+1} \left( 2rU_{2n-1}(r) - U_{2n-2}(r) \right) = -h^{n+1} U_{2n}(r).$$

This concludes the claimed identity for the even case. Finally, we show that the $2n + 1$ case follows from the $2n$ case:

$$H^{2n+1} = h^n \begin{bmatrix} 2r^2 & -r^2 \\ 1 & 0 \end{bmatrix} \begin{bmatrix} U_{2n}(r) & -rU_{2n-1}(r) \\ \frac{U_{2n-1}(r)}{r} & -U_{2n-2}(r) \end{bmatrix},$$

so that

$$[H^{2n+1}]_{11} = h^n \left( 2r^2 U_{2n}(r) - rU_{2n-1}(r) \right) = h^n \cdot rU_{2n+1}(r),$$

$$[H^{2n+1}]_{12} = -h^n \left( 2r^3 U_{2n-1}(r) - r^2 U_{2n-2}(r) \right) = -h^n \cdot hU_{2n}(r),$$

$$[H^{2n+1}]_{21} = U_{2n}(r),$$

$$[H^{2n+1}]_{22} = -rU_{2n-1}(r).$$

This completes the proof of the odd case, hence Lemma 15. ∎

To finish the proof of Lemma 14, we use the classical fact that $|U_j(r)| \leq j + 1$ for all $|r| \leq 1$, and note that each entry of $H^j$ is the value of some $U$, times a scalar between $-1$ and $1$; the $1/r$ factor in $[H^{2n}]_{21}$ gets absorbed because $h/r = r \leq 1$. This shows that for all $j \geq 2$, each entry of the $2 \times 2$ matrix $H^j$ has absolute value bounded by $|U_{j+1}(r)| \leq j + 1$; the same can be verified manually for $j = 1$. We conclude Lemma 14 by bounding $\|H^j\|_2 \leq \|H^j\|_1 \leq 2(j + 1)$. ∎

# C Proofs of the main theorems

In this section we use the potential-bounded regret and stability lemmas to complete the proofs of Theorems 7, 10, and 13.

*Proof of Theorem 7.* Substituting the result of Lemmas 5 and 6 in Theorem 4 gives the following:

$$\mathbb{E}[F(w_{\text{out}})] - F(w^*) \leq \frac{2\eta K \rho^2}{B} + \frac{\|w_{\text{init}} - w^*\|^2}{2\eta K T}.$$

Plugging in the choice of $\eta$ concludes the result. ∎

*Proof of Theorem 10.* Substituting the result of Lemmas 8 and 9 in Theorem 4 gives the following

$$\mathbb{E}[F(w_{\text{out}})] - F(w^*) \leq \frac{2\rho^2}{B\gamma}\left(1 - (1 - \eta\gamma)^K\right) + \frac{\gamma\|w_{\text{init}} - w^*\|^2}{2T} + \frac{\|w_{\text{init}} - w^*\|^2}{2\eta K T}$$

$$\leq \sqrt{1 + \frac{1}{K}} \cdot \frac{2\rho\|w_{\text{init}} - w^*\|}{\sqrt{BT}} + \frac{\beta\|w_{\text{init}} - w^*\|^2}{2KT}.$$

Plugging in the choice of $\eta$ now concludes the result. ∎

*Proof of Theorem 13.* We will use the notation $\lambda_t, d_t$ to denote the $\lambda_{\text{out}}, d_{\text{out}}$ returned by $\mathcal{A}$ at iteration $t$ of Algorithm 1. From Lemma 11 we get that

$$(\lambda_t^2 - \lambda_t)(\bar{f}_{\boldsymbol{\xi}^{(t)}}(w_{t+1}) - \bar{f}_{\boldsymbol{\xi}^{(t)}}(w^*)) - (\lambda_{t-1}^2 - \lambda_{t-1})(\bar{f}_{\boldsymbol{\xi}^{(t)}}(w_{t-1}) - \bar{f}_{\boldsymbol{\xi}^{(t)}}(w^*))$$

$$\leq \frac{1}{2\eta}(\|w_t + \lambda_t d_t - w^*\|^2 - \|w_{t+1} + \lambda_{t+1} d_{t+1} - w^*\|^2).$$

Let $\mathbb{E}_t[\cdot]$ be the expectation conditioned with respect to the randomness up time $t$ (inclusive). We now get from the uniform stability of $\mathcal{A}$ that

$$\mathbb{E}[F(w_{t+1}) - F(w^*)] = \underset{t-1}{\mathbb{E}}\left[\underset{\boldsymbol{\xi}^{(t)}}{\mathbb{E}}[F(w_{t+1}) - F(w^*)]\right]$$

$$= \underset{t-1}{\mathbb{E}}\left[\underset{\boldsymbol{\xi}^{(t)}}{\mathbb{E}}[\bar{f}_{\boldsymbol{\xi}^{(t)}}(w_{t+1}) - \bar{f}_{\boldsymbol{\xi}^{(t)}}(w^*)]\right] + O\left(\frac{\eta\rho^2 K^2}{B}\right).$$

Using the above inequalities, appropriately scaling and summing over $t$ and noting that $\lambda_0 = 1$ we get

$$\lambda_{T-1}^2 \left(\mathbb{E}[F(w_T)] - F(w^*)\right) = \frac{\|w_0 - w^*\|^2}{2\eta} + O\left(\frac{\eta\rho^2 K^2 \sum_t \lambda_t^2}{B}\right).$$

Using standard bounds on $\lambda_t$, we get that $\lambda_t = \Theta(tK)$. Substituting this in the above equation gives

$$\mathbb{E}[F(w_T)] - F(w^*) = O\left(\frac{\|w_0 - w^*\|^2}{2\eta K^2 T^2} + \frac{\eta\rho^2 K^2 T}{B}\right).$$

Now, using the value of $\eta$ prescribed in the theorem, we conclude the result. ∎

# D Experiment Details

## D.1 Datasets

**CoverType.** We used the scaled binary classification version of this dataset, as provided as a benchmark alongside `libsvm`. This dataset contains 581012 labeled examples, with feature dimension 54; thus, the logistic regression model has 110 parameters (including biases). Since this work is not concerned with generalization on holdout validation data, and this dataset does not come with a canonical train/test split, we trained on all of the examples. However, we note that logistic regression underfits to this dataset; the generalization gap was negligible when we tried random 90%-10% splits, and did not affect the trends seen in Figure 2.

**MNIST.** We used the training set of MNIST, which contains 60000 examples. The feature dimension is 764, and there are 10 classes, for a total of 7650 parameters (including biases). The pixels were normalized to the range $[0, 1]$. Again, the generalization gap is negligible in this setting; the results do not change (and the specific convergence times change only slightly) upon computing the convergence criterion using the canonical holdout validation set of 10000 examples.

In all experiments, batches were sampled with replacement (rather than the usual per-epoch shuffling convention), to remove artifacts arising from non-independence.

## D.2 Measuring convergence time

Thresholds for convergence were chosen to lie within 1% of the globally optimal training loss. We used 0.54 for CoverType and 0.3 for MNIST. We remark that although these choices are arbitrary, the trends exhibited in our experiments were not sensitive to the precise choice of threshold (although the convergence times can be dramatically different). To reduce variance, we record convergence when the mean of the past 10 losses lies below the threshold. Again, the trends in our experiments were not sensitive to this choice of aggregation. The means and standard deviations of convergence times in Figures 2 and 3 were computed over 20 runs.

## D.3 Hyperparameters

Learning rates were selected by grid search over an exponential grid (i.e. `numpy.logspace`) between 0.01 and 10, where consecutive candidates were $10^{1/20}$ apart.

The logistic regression models were trained with bias parameters; all parameters were initialized at zero.

## D.4 Computing infrastructure

To enable rapid evaluation of training losses on these ~100MB datasets, all optimization experiments were implemented in PyTorch on an NVIDIA V100 GPU machine. Each individual run took less than 1 minute.