[Reviews · NeurIPS 2020]

Review 1

Summary and Contributions: This paper provides the first convergence analysis of data echoing-based extensions of SGD, proximal SGD, and Nesterov's accelerated SGD, in which the corresponding algorithm takes additional gradient steps with the current minibatch before a new minibatch is arriving; hence the term "echoing". It focuses on the convergence and generalization for the described echoed gradient methods in convex optimization, and provides an understanding of bias-variance trade-off as batch sizes grows in the data echoing regimes.

Strengths: - Data echoing is a simple strategy that increases hardware utilization when the training pipeline has a bottleneck of data loading and processing. Previous empirical experiments already demonstrate the superiority of data echoing in large-scale machine learning problems but lacks theoretical support for this strategy. This work takes the first step to provide a theoretical understanding of convergence guarantees for several common data-echoed algorithms. - The data echoing analysis in this work characterizes the inherent bias-variance tension well, i.e., the optimization gains from repeating gradient updates versus the degrade of generalization from the overfitting to the persistent batches. It is interesting to see that the bias term is related to the echoing steps K while the variance term is related to the minibatch size B. As a result, the convergence results for several different SGD variants nicely reveals this property in the data-echoing setting. - The analytical framework in this work is very generic, and is extensible to other gradient based methods in the data-echoing regime. It provides a main data echoing theorem, which can be applied to certain specific algorithm by considering its stability and potential bounded regret properties.

Weaknesses: As is mentioned in the paper, one of the major applications of the data echoing algorithms is the massively parallel training of deep neural networks, which is non-convex. The current theoretical analysis and empirical study are only restricted to convex case. Although it is challenging to deal with the general non-convex case, I wonder if it is possible to derive any result with respect to some simple shallow networks, or some certain class of non-convex functions with bounded degree of nonconvexity. This would make the work even more interesting and contribute more to the community.

Correctness: The results make sense and seem to be correct, but I did not go through the proof details in the supplemental.

Clarity: The paper writing is very good, but I find several small problems related to notations, which could make confusion: - Between line 108-109, the authors use both the \bf\xi with a supscript "t" and the \bf\xi without a supscript "t", I guess for the latter the authors mean a general batch of samples does not depend on "t", but it is not explained clearly. Also, sometimes it has "i" in the supscript while othertimes it has "i" in the subscript. - In the Definition 1, line 120-121, I understand that \bf\xi and \bf\xi' are two batches of data differing in exactly one example, and the superemum is taken over all \xi, which is independent of \bf\xi or \bf\xi'. However, the reuse of the same notation really makes me confused for a while since it looks like \xi is some element belong to \bf\xi or \bf\xi'. - In Algorithm 1, I wonder why the output is an arithmetic average of all the w_t's? Is this a proof artifact? It makes more sense that if we want to do an averaging here, the w_t's should better have different weights such that the recent updates get higher score. - In Algorithm 1 line 4, we update w_{t} to w_{t+1} by executing algorithm A for K steps. However, in the equation below line 131, the subscript for w changes for each of those K steps. All these inconsistencies downgrade the readability of the paper. - Line 140 is a duplicate of line 132, which should be a typo here.

Relation to Prior Work: It clearly discussed the difference of this work from the previous one whose focus is empirical study only while this paper provides the theoretical convergence analysis.

Reproducibility: Yes

Additional Feedback: ********************* After rebuttal *************************** The authors' response has addressed my concerns, and I prefer to keep my original score. ******************************************************************


Review 2

Summary and Contributions: This papaer provides a convergence and generalization analysis of a model problem of data echoing-based gradient extensions of convex optimization methods. ======================== I have read the author's rebuttals, and the authors did not really address my concerns during rebuttal. I slightly raise my rating.

Strengths: 1. The topic under study, i.e., theoretical analysis of optimization with echoed gradient, is of general interesting. 2. The paper is overall well-written.

Weaknesses: 1. If the stochastic gradient in the echoed gradient-based gradient is computed with the same batch of training data? If so, this raises the efficiency issue of the use of training data; in particular, when K is large. I would expect that in each gradient computation step, randomly sample a large portion of these K examples would give better performance. This issue should be addressed in the paper. 2. Does Stochastic AGD converge (Table 2)? This result seems wrong. It is a well-known result in the first-order optimization community that any first-order optimization algorithm will accumulate error, in stochastic gradient, in SGD. 3. The experiments are too simple to verify the advantage of the echoed gradient-based SGD. The comparison of state-of-the-art algorithms, e.g. ADAM, SGD with momentum, etc., should be provided. Moreover, the experiments should be extended to large-scale experiments. 4. Can the analysis be generalized to nonconvex optimization? The most used machine learning algorithms are based on deep neural networks, which corresponds to highly nonconvex optimization problems. Convergence of echoed gradient based SGD for nonconvex optimization should be included in this paper.

Correctness: The convergence of stochastic AGD (in Table 2) seems incorrect, and the authors should verify this.

Clarity: The paper is well-written and easy to follow.

Relation to Prior Work: The paper does a good job in the literature review.

Reproducibility: Yes

Additional Feedback: Please see the weaknesses section.


Review 3

Summary and Contributions: This paper investigates data echoing on some simple gradient-based methods; namely, SGD, SGD + Nesterov acceleration, and prox SGD. In particular it investigates the bias/variance tradeoffs of extra passes through data for assisting with communication lag.

Strengths: The idea is well-motivated, and the structure / type of results are clean and useful. I can't speak too much on previous work; I have seen data echoing in numerical papers before, and believe there has been some theoretical work, but I haven't seen any on these specific methods.

Weaknesses: There is a lot of carelessness in the proofs. I think the claims are not outlandish, and the general proof structure seems reasonable, but the paper and the math seem unconnected at times. If not for that I would raise my score.

Correctness: - The citation [17] for the accelerated method is not correct. It refers to Lan's method, not Nesterov's acceleration, which are very different methods. Appendix: - line 322 What is equation 4, 6? same with line 323 Lemma 8 proof: Lambda, gamma, and eta included. Which is step size used? Also for prox, it should be the gradient at w_{t+1}. Lemma 11: indexing seems off, since d_t involves w_{t-1}, not w_{t+1} - line 351: do you mean gradient? What are the proofs in appendix B for? They seem somewhat unrelated and not used anywhere.

Clarity: Yes, for the most part the paper is well-written Figures 2,3, its a bit hard to figure out what I should be looking at. I assume that red line going down means data echoing has less overhead when you do it less?

Relation to Prior Work: I have seen previous works on data echoing, but those have mostly been numerical. There is most likely overlap between this work and stochastic optimization in general, e.g. the lan paper actually cited, where controlling the variance term is done by sampling more gradients. This should be discussed more.

Reproducibility: No

Additional Feedback: After rebuttal: I have read the author's rebuttal. While the contributions are interesting (resubmission encouraged), the presentation of the proofs were such that it's not easy to be 100% certain they are error-resistant. Additionally, the comment about the superfluous proofs (lemmas 14,15) were not addressed. Overall, I will keep my borderline score.

[Author Response · NeurIPS 2020]

We thank the reviewers for their comments. We first discuss some overall concerns raised by multiple reviewers, then
proceed to more specific points.

• *Results for non-convex*: Some reviewers mention the lack of results for non-convex settings as a weakness of the
paper. We highlight that to the best of our knowledge, our work is the first to provide precise theoretical regimes in
which data-echoing could lead to significant advantage. Indeed, if mere convergence results are expected, it is easy to
show that data-echoing (with an appropriate learning rate) converges in the non-convex case. Providing the precise
characterization of the benefits of data-echoing in non-convex settings is indeed very exciting future work. We believe
that the convex results provided in our paper will eventually provide the foundation for the non-convex results, as they
have in any other field of optimization.
• *Paper structure*: Reviewer 4 raised the concern that "the paper and the math seem unconnected at times". We disagree
with this remark. To clarify the general structure and role of the proofs (as is already laid out in the paper):
– 3 algorithms are analyzed (Thms. 7,10,13). The proof for each has the same structure viz. proving potential
bounded regret (Def. 3) and stability (Def. 1). Both properties are needed. This common structure is laid out in
Thm. 4.
– Lemmas 5, 8, 11 prove the potential bounded property. Due to similarity the proofs are bundled in Appendix A.
– Lemmas 6, 9, 12 prove the stability. Due to similarity the proofs are bundled in Appendix B.
– Finally the proofs of the main theorems are bundled in Appendix C. There are no other theorems/lemmas in the
paper.

**Reviewer 2:**   We appreciate the positive feedback, and the pointer to confusing notation. As you note, the notations
are not incorrect but could be confusing, and this will be addressed promptly in a revision.

• Average iterate as opposed to last iterate is indeed done to produce a simple and generalizable analysis. We believe
that akin to SGD (using techniques as in [Shamir & Zhang '12]), last iterate guarantees can be obtained here.

**Reviewer 3:**

• *Resampling batches*: Data-echoing is not being proposed as a general alternative to stochastic optimization methods.
If batches can be sampled at a rate compatible with the computation of gradients, then one should resample at every
iteration. **Data-echoing is relevant when batches cannot be sampled as fast** and we highlight the regime when it
could be advantageous over the current practice (of doing nothing).
• *Stochastic AGD will not converge*: The reference given in the table (to Lan's AC-SA) provides an accelerated method
for smooth stochastic optimization with the rate mentioned in the table. We disagree with the "well-known result in
first-order optimization community" that contradicts this. If your objection is that Nesterov's acceleration does not
directly work for SGD, as noted by Reviewer 4, then this will be clarified.
• *Comparisons with Adam, etc.*: This is moot with respect to the scope of the paper. Such comparison requires
developing principled data-echoed variants of adaptive methods. A first attempt at this comparison was made in [Choi
et al. '19]. Here we focus on providing a theoretical foundation for data-echoing. Data-echoed adaptive methods are
interesting for future investigation.

**Reviewer 4:**   We thank the reviewer for reading closely and pointing out typos. We stress that the issues pointed out
are mere typos, as we highlight below. (We provided a detailed discussion regarding the point of "unconnected math"
earlier.) We request the reviewer to revisit these and consider increasing their score, post clarifications.

• *Nesterov Acc vs Lan's AC-SA*: Thanks for pointing out this nuance, which might confuse other readers. We will clarify
in the final version.
• *Unrelated appendix*: Appendix B contains stability proofs (for Lem 6,9,12). Lemmas are at lines 168,179,204 in
the main paper and are necessary for the main proofs (Thms 7,10,13). It is clearly stated that proofs appear in the
appendix.
• *Related work*: Resampling of batches as a trick to *reduce variance* is of course ubiquitous in literature (equivalent to
increasing batch size). Performing *multiple gradient steps on the same batch* has not been analyzed in stochastic
optimization as far as we know. Nevertheless, we request that the reviewer provide precise references, and we will
include and discuss them.
• *Typos*:
– line 322-323: These are misplaced references: (4) and (6) should read (2) and (3).
– Lemma 8: The only typo we spot here is Line 324 should say $\gamma$ strong convexity instead of $\lambda$. The step-size is
clearly defined in the description of the algorithm in Line 137.
– Lemma 11: There is no indexing error as far as we can see. There is however a minor typo below Line 336, that
might be the source of confusion. Corrected version below:

$$\lambda_t \|\eta \nabla f(x_{t+1})\|^2 - 2\eta \nabla f(x_{t+1})^\top (w_t + \lambda d_t - w) = \lambda_t^{-1} \left( \|w_t + \lambda_t d_t - w - \lambda_t \eta \nabla f(x_{t+1})\|^2 - \|w_t + \lambda_t d_t - w\|^2 \right).$$

– Line 351: Indeed, RHS is the difference between gradients. Thanks.

[Meta-Review · NeurIPS 2020]

The paper is a theoretical analysis of the behaviour of "echoed gradients" in convex optimization. The investigation is timely, and will cast light on an interesting area of current practice. More than one reviewer believes the paper should explicitly handle the non-convex case. I disagree, and side with the authors that the convex case is sufficient. The relevant non-convex optimizers generally contain convex stepping as a subprogram, so this analysis is reasonable.  A point where I do side strongly with the reviewers is the topic of carelessness in presentation. All agree the paper is well *written*, but that there are errors in the math. The authors rebuttal dismisses these complaints as mere "typos" or "nuance". I find this rather unconvincing. Typos in math are considerably harder on the reader than typos in prose, and the tone of the rebuttal seemed to imply it was the reviewers' fault for not autocorrecting the typos rather than the authors' for including them.